# ATIP1 Is a Suppressor of Cardiac Hypertrophy and Modulates AT2-Dependent Signaling in Cardiac Myocytes

**DOI:** 10.3390/cells14090645

**Published:** 2025-04-28

**Authors:** Tobias Fischer, Sina Gredy, Nadine Scheel, Peter M. Benz, Benjamin Fissler, Melanie Ullrich, Marco Abeßer, Adam G. Rokita, Jochen Reichle, Lars S. Maier, Oliver Ritter, Hideo A. Baba, Kai Schuh

**Affiliations:** 1Institute of Physiology I, University of Wuerzburg, 97070 Wuerzburg, Germanysina.kollert@uni-wuerzburg.de (S.G.);; 2Department of Cardiology and Pneumology, Georg-August-University, 37075 Goettingen, Germany; 3Department of Internal Medicine II, University Clinic Regensburg, 93053 Regensburg, Germany; 4Brandenburg University Hospital, Department of Internal Medicine I, 14770 Brandenburg an der Havel, Germany; 5Institute of Pathology, University of Duisburg–Essen, 45147 Essen, Germany

**Keywords:** ATIP1, MTUS1, knockout mice, cardiac hypertrophy, angiotensin-type-2 receptor

## Abstract

So far, the molecular functions of the angiotensin-type-2 receptor (AT2) interacting protein (ATIP1) have remained unclear, although expression studies have revealed high levels of ATIP1 in the heart. To unravel its physiological function, we investigated ATIP1-KO mice. They develop a spontaneous cardiac hypertrophy with a significantly increased heart/bodyweight ratio, enlarged cardiomyocyte diameters, and augmented myocardial fibrosis. Hemodynamic measurements revealed an increased ejection fraction (EF) in untreated ATIP1-KO mice, and reduced end-systolic and end-diastolic volumes (ESV and EDV), which, in sum, reflect a compensated concentric cardiac hypertrophy. Importantly, no significant differences in blood pressure (BP) were observed. Chronic angiotensin II (AngII) infusion resulted in increases in BP and EF in ATIP1-KO and WT mice. Reductions in ESV and EDV occurred in both ATIP1-KO and WT but to a lesser extent in ATIP1-KOs. Isolated cardiomyocytes exhibited a significantly increased contractility in ATIP1-KO and accelerated Ca^2+^ decay. AngII treatment resulted in increased fractional shortening in WT but decreased shortening in ATIP1-KO, accompanied by accelerated cell relaxation in WT but absent effects on relaxation in ATIP1-KO cells. The AT2 agonist CGP42112A increased shortening in WT cardiomyocytes but, again, did not affect shortening in ATIP1-KO cells. Relaxation was accelerated by CGP42112A in WT but was unaffected in ATIP1-KO cells. We show that ATIP1 deficiency results in spontaneous cardiac hypertrophy in vivo and that ATIP1 is a downstream signal in the AT2 pathway regulating cell contractility. We hypothesize that the latter effect is because of a disinhibition of the AT1 pathway by impaired AT2 signaling.

## 1. Introduction

The renin–angiotensin system (RAS) comprises a well-characterized cascade of proteases and hormones involved in the regulation of various physiological processes, and the dysregulation of this cascade plays a critical role in widespread diseases, like hypertension, cardiac hypertrophy, and heart failure [1,2,3]. The cellular effects of the end-product angiotensin II (AngII) are mainly exerted by two subtypes of AngII receptors. AT1 is involved in most of the well-established physiological effects of AngII, and AT2 was shown to counterbalance some of the effects of AT1 activation, although the role of the latter is still uncertain [1,4,5]. Both receptors belong to the superfamily of G-protein-coupled receptors but exert different signals and functions [6]. AT1 activates growth pathways and mediates major AngII effects, such as vasoconstriction, increased cardiac contractility, vascular and cardiac hypertrophies, inflammatory responses, and oxidative stress. The activation of AT2 is believed to counteract AT1’s effects, including vasodilation, antigrowth, and anti-hypertrophic effects [1,6,7,8,9,10], and it is involved in blood pressure regulation [11]. In addition to coupling to G-proteins [12,13], AT2 directly activates SH2-domain-containing phosphatase 1 (SHP-1) by a mechanism distinct from that of the classical paradigm of heterotrimeric G-proteins [14]. Interaction with SHP-1 is supposedly related to the anti-proliferative effects of AT2 [5,15]. On the contrary, it was shown that the recruitment of the promyelocytic zinc finger protein PLZF to AT2 is associated with increased protein synthesis in cardiomyocytes and cardiac hypertrophy [16]. In sum, the AT2-dependent signaling pathway is incompletely understood [4,5].

Initially described as human *mitochondrial tumor suppressor gene 1* (MTUS1) [17], MTUS1 has not only been shown to act as a potential tumor suppressor in a variety of cancers but also to operate as an interaction partner of AT2, linking the AT2 pathway to carcinogenesis [18]. Because of that, MTUS1 has also been named AT2-interacting protein (ATIP) [19,20] and AT2-binding protein (ATBP) [21]. For reasons of clarity, we have used the ATIP1 nomenclature throughout this study.

ATIP1 is involved in the transport of AT2 from the Golgi compartment to the plasma membrane and mediates inhibitory effects on MAP kinases and anti-proliferative effects of AT2 [21]. Because ATIP1 trans-inactivates receptor tyrosine kinases, it was considered as an early component of growth-inhibitory-signaling cascades [20]. Poly (ADP–ribose) polymerase-1 (PARP-1) activates the transcription of the MTUS1 gene and represses AT2 gene transcription [22]. PARP-1 is involved in regulating blood pressure and inflammatory processes, but PARP-1-deficient mice do not develop cardiac hypertrophy [23]. Mice ubiquitously overexpressing ATIP1 showed attenuated superoxide anion production, the activation of cell proliferative signaling cascades, and the elevated expression of tumor necrosis factor α [24]. After femoral artery cuff placement, neointima formation was reduced in these mice, suggesting a role in vascular remodeling [24].

The first indications toward an anti-hypertrophic function of ATIP1 in the myocardium arose from loss-of-function mouse model studies. However, in the first descriptive study, no underlying mechanisms were investigated [25], and in a second transgenic model, the overexpression of a splice variant resulted in the suppression of cardiac hypertrophy in response to pressure overload and phenylephrine treatment [26]. More recently, the discovery of a de novo mutation in the human ATIP1 gene added the clinical aspect that wild-type ATIP1 decreases the risk of non-compaction of the ventricular myocardium [27].

Herein, we intend to address the cardiac functions of ATIP1 in vivo, and we investigated the underlying molecular regulatory mechanisms. We used a gene-trap-based ATIP1 knockout mouse model and analyzed cardiac functions in vivo by hemodynamic analyses, cardiac myocyte function ex vivo, and molecular changes in isolated organs and cells.

## 2. Materials and Methods

### 2.1. Generation of ATIP1-KO Mice

ATIP1-KO mice were generated via a gene trap approach [28,29]. In brief, the embryonic stem cell line RRA048 (International Gene Trap Consortium) was used for blastocyst injection and subsequent foster mother implantation to generate germ line chimeras. The RRA048 cells contain the pGT1Lxf gene trap vector, composed of a part of the engrailed 2 intron 1 (en2 intron 1); a splice acceptor (*SA*); a β-galactosidase/neomycin-resistance fusion gene (*β*-geo), required for selection and X-Gal staining; an SV40 polyadenylation sequence (*SV40pA*); and a pUC backbone vector sequence. According to our RACE-PCR results (see below), yielding only the mATIP1 isoform 4, this vector was inserted between corresponding exons 3 and 4 of the Mtus1 gene. Throughout this study, adult ATIP1-KO mice (6–10 months of age), with a biallelic knockout of ATIP1, and wild-type littermate control mice were used. To avoid inbreeding effects resulting from the genetic background, we used mice with a mixed 129/Ola × C57Bl/6 genetic background. All the experiments and animal care were performed in accordance with German legislation on the protection of animals and the guide for the care and use of laboratory animals and were approved by the local governmental animal care committee of the Government of Lower Franconia, Bavaria, Germany (approval code: 55.22531.01-62/08). The offspring were genotyped using the following sets of primers: for WT-PCR, amplifying a 900 bp fragment, primer 1 (located in intron 9 in the forward orientation) 5′-CTATGGCAATAAGACAGCCC-3′ and primer 3 (located in intron 9 in the reverse orientation) 5′-GGTTTGATCCCCAACACCAC-3′, and for KO-PCR, amplifying a 500 bp fragment, primer 1 and primer 2 (located in the gene trap vector in the reverse orientation) 5′-CACTCCAACCTCCGCAAACTC-3′.

### 2.2. Cloning of Mouse ATIP1 Heart-Specific Splice Variants by RACE-PCR

ATIP1 splice variants were detected by the rapid amplification of cDNA ends (the RACE method) from a mouse heart (Marathon-Ready™ double-stranded cDNA library, Clontech), using the Advantage^®^ 2 PCR system (Clontech) and the following gene-specific and adaptor primers: mouse ATIP1 forward (R1), 5′-GGCAGGCAGAAGAATCCCAG-3′; mouse ATIP1 reverse (R2), 5′-AGGTGCTGGATCACAACGG-3′; and adaptor primer (AP), 5′–CCATCCTAATACGACTCACTATAGGGC–3′. The resulting PCR fragments were cloned in pCR^®^-XL-TOPO (Invitrogen) and subsequently sequenced with standard M13 uni (−21) and M13 rev (−29) primers. The DNA sequences were analyzed and aligned with DNAMAN 4.0 (Lynnon BioSoft).

### 2.3. RT-PCR and Northern Blot Analyses

The total RNA was extracted from ATIP1-KO and WT mouse hearts, using 1 mL of TRIzol reagent (Invitrogen) for 50 mg of tissue, and RNA was isolated according to the manufacturer’s instructions. For RT-PCR analyses, 0.5 µg of the total RNA was used with the OneStep RT-PCR Kit (Qiagen). Variant-specific primers were used, including isoform 1, 5′-CAGGTACAGACGTCTCCAAGG-3′; isoform 3, 5′-GCATGAGGGATGCATGAACATG-3′; and isoform 4, 5′-GGCATGCACAGTGTGATGTAAT-3′, in combination with the common reverse primer, 5′-GTACGTAGCTGAAACGGTTGG-3′. To test for RNA integrity, a GAPDH fragment was amplified in parallel using the following primers: GAPDH forward, 5′-ACCACAGTCCATGCCATCAC-3′, and GAPDH reverse, 5′-TCCACCACCCTGTTGCTGTA-3′. For northern blot analyses, we used a standardized MessageMap northern blot membrane with 2 μg of polyA RNA (Stratagene; BALB/c male mouse) per lane. The membrane was hybridized with a Biotin-16-dUTP-labeled (Roche) Mtus1 cDNA probe. Hybridization signals were detected using streptavidin–horseradish peroxidase (a chemiluminescent nucleic acid detection module; PIERCE) on X-ray films.

### 2.4. Western Blotting

To obtain lysates of the mouse organs, 50 mg of tissues were homogenized with an UltraTurrax (IKA) in 1 mL of PBS/2% SDS, supplemented with complete protease and PhosSTOP phosphatase inhibitor cocktails (Roche Diagnostics, Germany). The proteins were separated using 10 or 12% SDS-PAGE under reducing conditions and electro-transferred to nitrocellulose membranes (Schleicher & Schuell Protran, Fisher Scientific, Hampton, USA), using semi-dry blotters (Biometra, Goettingen, Germany). The polyclonal rabbit anti-ATIP1 antibody was established by Eurogentec (Seraing, Belgium). Therefore, the peptide H2N-CPRNSGSFPSPSISPR-COOH (amino acids 422–436 of the human ATIP1 protein sequence) was coupled to hemocyanin. Rabbits were immunized with the modified peptide on days 0, 14, 28, and 56, and the final bleed was carried out at day 80 of the immunization procedure. Subsequently, the rabbit serum was affinity purified with the antigen peptide coupled to an AF–Amino Toyopearl 650 M matrix (Tosoh Bioscience, Stuttgart, Germany). The blots were probed using anti-ATIP1, rabbit anti-p44/42 MAP kinase (9102, Cell Signaling), rabbit anti-phospho-p44/42 MAP kinase (9101, Cell Signaling), rabbit anti-GAPDH (14C10), rabbit anti-Cav1.2 (ACC-003, Alomone Labs), monoclonal mouse anti-PLB A1 (A010-14, Badrilla), monoclonal mouse anti-cardiac actin (A9357, Sigma, St. Louis, MO, USA), polyclonal rabbit anti-SERCA2a (A010-20, Badrilla, Leeds, UK), and monoclonal mouse anti-CaMKII (MA1-047, Pierce, Thermo Fisher Scientific, Hampton, NH, USA) antibodies, followed by an anti-rabbit or anti-mouse horseradish-peroxidase-conjugated secondary antibody (TrueBlot^®^, eBioscience, San Diego, CA, USA), all diluted in 5% non-fat milk/PBS supplemented with 0.05% Tween 20. For signal detection, the ECL Plus reagent (Amersham, Freiburg, Germany) was used.

### 2.5. Quantification of ERK Phosphorylation and Protein Expression Levels

The western blot band intensities were quantified using ImageJ (NIH, MD, USA). The data were calculated as the ratio of arbitrary densitometric units either of P-ERK to the total ERK or of the protein signal’s intensity in relation to the actin or GAPDH signal’s intensity.

### 2.6. Immunohistochemistry

The dissected hearts were embedded in Tissue-Tek (Sakura, Umkirch, Germany) and snap-frozen in N_2_. Then, 5–10 µm cryosections were cut using an HM500OM microtome (MICROM, Neuss, Germany) and collected on SuperFrost slides (Thermo Fisher Scientific, Hampton, USA). The cardiomyocytes were isolated using Liberase/trypsin digestion according to Protocol PP00000125 of the Alliance for Cellular Signaling (AfCS, doi: https://doi.org/10.1038/nature01304). Tissue slices and isolated cardiomyocytes (described below) were fixed with 4% paraformaldehyde/PBS for 10 min, permeabilized with 0.2% TritonX-100/PBS for 20 min, and blocked in 5% goat serum/PBS for 1 h. Overnight incubation with primary rabbit polyclonal anti-ATIP1 (1:500; Eurogentec, Seraing, Belgium) and mouse monoclonal anti-plasma membrane Ca^2+^ ATPase (5F10, Sigma, St. Louis, MO, USA) was followed by incubation with secondary goat–anti-rabbit and goat–anti-mouse antibodies conjugated to Alexa Fluor 488 or 594 for 2 h (Thermo Fisher Scientific, Hampton, USA) or Phalloidin–TRITC (P1951, Sigma, St. Louis, MO, USA) to stain the cardiac actin. For AT2/ATIP1 co-localization, the direct-coupled rabbit polyclonal anti-AT2-ATTO-488 (AAR-012-AG, Alomone Labs) antibody was used after labeling the anti-ATIP1 antibody with the secondary Alexa Fluor 594 antibody. All the antibody incubations were performed in 5% goat serum/PBS in a humidified chamber in the dark. Stained sections were investigated using an Eclipse E600 microscope equipped with a C1 confocal scanning head (Nikon, Duesseldorf, Germany).

### 2.7. Cell Culture and Immunoprecipitations

HEK 293 cells were cultured as previously described [30], grown to ~80% confluence, and transfected using LipofectamineTM 2000 (Invitrogen) according to the manufacturer’s instructions. After 48 h, the cells were rinsed with PBS and lysed in RIPA buffer without SDS. Subsequent to 10 min of incubation on ice, the cells were scraped from the plates and pelleted at 13000 g for 10 min at 4 °C. Then, 300 µg of the lysate was used for each immunoprecipitation, and 3 µg of affinity-purified anti-ATIP1 antibodies or 3 µg of anti-AT2 antibody (Santa Cruz, SC-9040, Heidelberg, Germany) was added. After 1 h of rocking at 4 °C, 50 µL of protein A/G agarose was added to the mixtures and rotated overnight at 4 °C. The protein A/G agarose was pelleted at 4000 g for 30 s and washed twice with RIPA buffer containing protease inhibitors. The bound proteins were eluted in 2 × Laemmli buffer, separated on polyacrylamide gels, and transferred to nitrocellulose for western blotting, as described above.

### 2.8. X-Gal Staining

X-Gal staining was performed as previously described [29,31].

### 2.9. Histology, Morphometrical Analyses, and Cardiac Fibrosis

For histological analysis, hearts were fixed in 4% formaldehyde and embedded in paraffin. Then, 5 μm sections were stained with periodic acid–Schiff reagent or 0.1% picrosirius red. Photomicrographs of myocardial sections were photographed with a CCD-FV2T camera (Olympus) and analyzed with the CellD image analysis system (Olympus), with the investigator blinded to the genotypes. The mean cardiomyocyte diameters were calculated by measuring the diameter of 100 cells per section at the level of a centrally located nucleus. Sections were stained with picrosirius red to detect collagen deposition to assess fibrosis. The level of fibrosis was calculated as a percentage of myocardial fibrosis detected according to the method described in the literature [32,33]. Myosin-heavy-chain expression was quantified as described before [34].

### 2.10. Isolation of Cardiomyocytes

Using a Langendorff perfusion apparatus, cardiomyocyte isolation from mouse hearts was performed as described previously [35,36]. In brief, the mice were anesthetized in a gas chamber with isoflurane, and the hearts were excised and mounted on a Langendorff perfusion system and perfused with a nominally Ca^2+^-free Tyrode’s solution containing (in mM) NaCl 113, KCl 4.7, KH_2_PO_4_ 0.6, Na_2_HPO_4_ × 2H_2_O 0.6, MgSO_4_ × 7H_2_O 1.2, NaHCO_3_ 12, KHCO_3_ 10, HEPES 10, taurine 30, BDM 10, glucose 5.5, and phenol red 0.032 for 4 min at 37 °C and pH 7.4. Then, the perfusion solution was switched to the same solution containing 7.5 mg/mL of Liberase 1 (Roche diagnostics, Mannheim, Germany), trypsin 0.6%, and 0.125 mM CaCl_2_. Perfusion was continued for about 3 min, until the heart became flaccid. The ventricular tissue was removed, cut into small pieces, and dispersed until no solid cardiac tissue was left. Ca^2+^ reintroduction was performed carefully via stepwise increasing the Ca^2+^ concentration from 0.1 mM to 0.8 mM. Shortly thereafter, the cells were plated on superfusion chambers, with the glass bottoms treated with Laminin to allow cell adhesion, and then used for immediate measurements.

### 2.11. Cardiomyocyte Shortening and Ca^2+^ Measurements

Cardiomyocyte shortening and Ca^2+^ measurements were performed as described previously [37,38]. In short, myocytes were loaded with Fluo-3 AM (10 µmol/L; Molecular Probes, Eugene, OR, USA) in darkness for 15 min prior to being mounted on an inverted microscope (Nikon Eclipse TE2000-U, Duesseldorf, Germany) connected to a fluorescence detection system (IonOptix, Amsterdam, The Netherlands). The emitted fluorescence was recorded using a photomultiplier at ~535 ± 20 nm. Ca^2+^ transient amplitudes were calculated as the peak fluorescence divided by the baseline fluorescence after the background fluorescence had been subtracted (F/F0). Ca^2+^ values were calculated using a semi-quantitative approach as [Ca^2+^]_i_=K_d_(F/F_0_)/(K_d_/[Ca^2+^]_i-rest_+1-F/F_0_)), with a Kd value of 1100 nmol/L and an estimated [Ca^2+^]_i-rest_ of 100 nmol/L, as described before [37]. In a subset of experiments, angiotensin II (AngII; 0.5 μmol/L; Calbiochem, San Diego, CA, USA) was added to the superfusion solution. Prior to starting the measurement recordings, the myocytes were superfused and stimulated at a basal rate of 0.5 Hz for 10 min to allow a complete wash-in of the superfusion solution and steady-state contractions. Similarly, in another subset, the selective AT2 agonist CGP42112A (10 μmol/L; Sigma–Aldrich, Steinheim, Germany) had been added to the superfusion solution 10 min before the measurements were started, again ensuring steady-state conditions.

### 2.12. Chronic Infusion of AngII

Seven-month-old ATIP1-KO and WT control mice received AngII (350 ng/kg BW/min, Sigma) for a period of 2 weeks. The substance was dissolved in 0.9% NaCl and then infused subcutaneously via osmotic mini pumps (model 2002; Alzet, Colorado City, CO, USA, n = 12 mice per group). All the mice were analyzed and sacrificed under isoflurane anesthesia.

### 2.13. Transversal Aortic Constriction (TAC)

Male mice with a bodyweight of 25–30 g underwent transversal aortic constriction (n = 12). The mice were anesthetized with isoflurane, and the transversal aorta was prepared. A 27-gauge needle was tied against the aorta between the carotid arteries, using a 6-0 non-absorbable suture. After the removal of the 27-gauge needle, the skin was closed, and the mice were kept under buprenorphine analgesia for 3 days.

### 2.14. Non-Invasive Blood Pressure Estimation, Hemodynamics, and Tissue Harvesting

The arterial blood pressure was measured after 7 days of acclimatization in ATIP1-KO and WT mice, using a tail cuff (Softron, Tokyo, Japan). After 2 weeks of chronic AngII treatment or TAC, respectively, the cardiac function was measured compared to basal recordings in untreated mice, using a pressure–volume conductance catheter (Ultra-Miniature SPR-839, Millar Instruments) connected to an MPVS Ultra (Millar Instruments) and a PowerLab 16/30 (ADInstruments) for data acquisition. Values were calculated with PVAN Ultra 1.1 data analysis software (Millar Instruments), and the analyses were performed as described previously [39]. All the mice were analyzed and sacrificed under isoflurane anesthesia, and the left ventricles were snap-frozen in liquid N_2_ for further analyses.

### 2.15. Statistical Analysis

The results are expressed as means ± SDs. The datasets were analyzed using two-sample *t*-tests and two-way ANOVA or Welch’s tests, depending on the homogeneity of the variances, and *p* values of <0.05 were considered as significant.

## 3. Results

### 3.1. Spatial ATIP1 Expression in Cardiac Myocytes and the Interaction of ATIP1 with AT2

ATIP1 localization in the heart and in isolated cardiomyocytes was investigated using confocal laser scanning microscopy. The overlay of the ATIP1 and AT2 staining demonstrated the co-localization of both proteins, most likely at the plasma membrane (Figure 1A–C). Confirmation of the sarcolemmal localization of ATIP1 was achieved by the co-staining of the plasma membrane’s Ca^2+^ ATPase (PMCA, Figure 1D–F). Additionally, the counterstaining of the total actin revealed no co-localization with cytoplasmic organelles or structures, e.g., actin, but suggested again the localization of ATIP1 at the plasma membrane (Figure 1G). Furthermore, northern blot analyses revealed strong expression of ATIP1 not only in the heart but additionally in the kidneys, liver, and testes (Figure 1J). To investigate the possible protein–protein interaction of ATIP1 with AT2, we transfected HEK293 cells with expression vectors, for ATIP1 or for AT2 or both, and immunoprecipitated them either with an anti-ATIP1 or an anti-AT2 antibody and detected, in both cases, the interaction of the two proteins, using an antibody for the potential interaction partner, respectively (Figure 1K,L).

### 3.2. Cloning of the Heart-Specific ATIP1 Variant

The appearance of a single band in the northern blots prompted us to elucidate which of the predicted ATIP1 splice variants might be expressed in the heart. Therefore, we cloned ATIP1 splice variants from a mouse heart cDNA library, using 5′ and 3′ RACE. The gene-specific primers R1 and R2 were located in an exon representing all three predicted splice variants (predicted in exon 8, Figure 2A). Surprisingly, only the murine ATIP variant 4, equal to ATBP50 and Mtus1 variant 4, GenBank NM_001005865.2, was detected. In addition, further RT-PCR experiments with splice-variant-specific primers gave no evidence for the appearance of additional variants and confirmed our findings obtained by RACE-PCR. In other words, mATIP1 variant 4 was the only detectable variant in the heart (Figure 2B).

### 3.3. ATIP Gene Trapping

To study the physiological functions of ATIP1 in vivo, we generated ATIP1-KO mice by blastocyst injection of the embryonic stem cell clone RRA048 (International Gene Trap Consortium). In these, the gene trap is inserted between the third and the fourth exons of the murine-heart-specific ATIP1 splice variant (Figure 2C). The resulting chimeric males were then used to produce ATIP1-null mice. These were viable and fertile and macroscopically indistinguishable from wild-type controls. The genotyping of the mice was performed using PCR analyses and a common forward primer (Pr. 1) and a reverse primer either in the artificial β-geo gene for the KO-PCR (Pr. 2) or a reverse primer in the intron downstream of the gene trap for the WT-PCR (Pr. 3, Figure 2C). The PCR products were about 500 bp for the KO allele and 900 bp for the WT allele (Figure 2D).

### 3.4. ATIP1 Deficiency at the RNA and Protein Levels

The successful disruption of the ATIP1 gene at the RNA level was assured by RT-PCR, using the total RNA of the organs obtained from the KO and WT littermates. Shown as an example, RT-PCR using knockout heart RNA gave no PCR product for ATIP1, as compared to glyceraldehyde-3-phosphate dehydrogenase (GAPDH, Figure 2E). Subsequent western blot analyses with polyclonal rabbit anti-ATIP1 antibodies, raised against the shared C-terminus, and AT2-specific antibodies confirmed ATIP1 deficiency at the protein level and unchanged AT2 expression in the hearts of the KO mice (Figure 2F).

### 3.5. ATIP1 Promoter Activity in Cardiac Myocytes

To obtain a more detailed impression of the promoter activity of the ATIP1 gene, we analyzed the activity of the artificial β-geo construct brought under the control of the natural ATIP1 gene promoter. The X-gal staining of the hearts from homozygous KO mice, of heart cryosections, and of isolated cardiomyocytes revealed easily detectable promoter activity in practically all the areas of the heart, including cardiac vessels, as well as in the cardiac myocytes of the atria, ventricles, and septum wall (Figure 3A,B,D,E,G). Negative controls were prepared from WT hearts (Figure 3C,F,H).

### 3.6. Cardiac Hypertrophy in ATIP1-KO Mice

The ATIP1-deficient mice developed a spontaneous, gender-independent cardiac hypertrophy, which was reflected by a significantly increased heart-to-bodyweight ratio of 6.4 mg/g as compared to 4.7 mg/g in WT animals (n = 20 of each genotype; *p* < 0.05, Figure 4A). Non-invasive blood pressure measurements demonstrated no significant changes in blood pressure in the ATIP1-KO mice (Figure 4B, n = 25 of each genotype). Morphometrical analyses of the cardiac myocytes showed significantly enhanced cardiomyocyte diameters in KO animals vs. WT controls (14.88 vs. 11.03 µm, *n* = 100 cardiac myocytes in sections of 12 mice of each genotype; *p* < 0.05, Figure 4C). In line with this, myocardial fibrosis was nearly doubled in KO animals (1.97 ± 0.09-fold relative content vs. wild-type, n = 12 mice of each genotype, *p* < 0.05, Figure 4D). Furthermore, we saw an increase in myosin-heavy-chain expression in KO mice as compared to that in WT controls (1.71 ± 0.48-fold expression vs. wild-type, n = 12 mice of each genotype, *p* < 0.05, Figure 4E). From these data, we concluded that cardiac hypertrophy in ATIP1-KO mice was not caused by pressure overload but might be caused by the missing mediation of the anti-hypertrophic AT2 signaling in ATIP1-deficient mice.

### 3.7. Increased Shortening and Reduced Responses to AngII and CGP42112A in ATIP1-Deficient Cardiomyocytes

In a first series of experiments, we investigated cardiomyocyte shortening and Ca^2+^ transient amplitudes under basal conditions and subsequently after treatment with AngII (Figure 5). In a second series of experiments, we investigated, again, cardiomyocyte shortening and Ca^2+^ transients under basal conditions and after treatment with the AT2 agonist CGP42112A (Figure 6). Under basal conditions, cardiomyocytes from ATIP1-KO mice revealed significantly increased cell shortening at low stimulating frequencies compared to those of the controls (e.g., at 0.5 Hz, 4.1 ± 0.5 vs. 2.4 ± 0.3%, n = 14 vs. 11, *p* < 0.05) and a strong trend toward increased contractility at higher frequencies, whereas the Ca^2+^ transient amplitudes did not differ. Treatment with AngII significantly increased cell shortening in cardiomyocytes from WT mice (at 1 Hz, 1.9 ± 0.2 vs. 3.5 ± 0.5%, *n* = 11 and 12, respectively, *p* < 0.05) but reduced it in ATIP1-KO cells (at 1 Hz, 3.1 ± 0.5 vs. 1.8 ± 0.2%, *n* = 13 and 15, respectively, *p* < 0.05), whereas the Ca^2+^ transient amplitudes remained, again, unchanged. The investigation of cell relaxation properties under basal conditions revealed a slightly faster cell relaxation at all the stimulation frequencies (with a significantly accelerated relaxation only at 0.5 Hz in ATIP1-KO vs. WT: 222 ± 30 vs. 337 ± 45 ms, n = 14 vs. 11, *p* < 0.05) and significantly accelerated the diastolic Ca^2+^ transient decline at all the stimulation frequencies in cells from ATIP1-KO mice compared to those from WT mice. Interestingly, AngII significantly accelerated both cell relaxation and Ca^2+^ decline in cardiomyocytes from WT mice, but it hardly affected relaxation in cells from ATIP1-KO mice (see Figure 5).

Stimulation with the AT2 agonist CGP42112A (Figure 6) increased cardiomyocyte shortening in WT cells significantly (e.g., at 1 Hz (in percentages): 3.0 ± 0.3 vs. 2.0 ± 0.3, n = 19 and 25 for WT and WT plus CGP, respectively, *p* < 0.05), but it did not alter shortening in cardiomyocytes for ATIP1-KO mice. Again, no significant differences were detected concerning Ca^2+^ transient amplitudes between these groups. Regarding relaxation properties, CGP42112A significantly accelerated both cell relaxation and Ca^2+^ decline in WT cells at all the measured frequencies compared to those measured under basal conditions (e.g., at 1 Hz, a relaxation time of 80% of the cell length before vs. after CGP42112A: 174 ± 15 vs. 107 ± 15 ms, n = 7 vs. 8, *p* < 0.05).

### 3.8. Altered Left-Ventricular Cardiac Function Under Basal Conditions and After Chronic AngII Application

Under basal conditions, most estimated left-ventricular volumes were reduced in KO mice, including the maximum volume, minimum volume, end-systolic and end-diastolic volumes, and volumes at dP/dt max and min (Table 1). Additionally, interpreted as the diastolic index, the relaxation constant Tau (W) was significantly lower in KO mice (Table 1). Taken together, these data pointed toward compensated concentric cardiac hypertrophy, including slower relaxation caused by a more rigid ventricular wall.

Low-dose (350 ng/kg/min) chronic AngII treatment for a time period of two weeks further accentuated the cardiac effects of ATIP deficiency in vivo. In addition to the parameters that were already altered under basal conditions, like the end-systolic and end-diastolic volumes and volumes at dP/dt max (e.g., the maximum volumes for WT (14.49 ± 2.7) vs. KO (11.63 ± 2.5), minimum volumes for WT (5.33 ± 1.7) vs. KO (3.67 ± 2.2), and end-diastolic volumes (EDVs) for WT (13.72 ± 2.7) vs. KO (10.42 ± 2.4 µL), Table 1), the cardiac outputs (COs for WT (4445.77 ± 1090.7) vs. KO (3519.55 ± 788.9 µL/min); n = 12 mice of each genotype; *p* < 0.05) were further reduced in the ATIP1-KO animals (Table 1). This further reduction in the left-ventricular performance after chronic AngII treatment obviously reflects the augmented response to AngII in KO mice, most likely mediated by a stimulation of pro-hypertrophic AT1 signaling and missing counter-acting AT2 downstream signaling.

### 3.9. Compared to Chronic AngII Administration, Pressure Overload by TAC Resulted in an Altered Hypertrophic Response in ATIP1-Deficient Mice

To verify if the changes in the cardiac performances of the ATIP1 mice after chronic AngII administration are AngII/AT1 dependent or simply reflect a common response to hypertrophic stimuli, we also measured hemodynamics after three weeks of pressure overload by transversal aortic constriction. Despite a similarly elevated ventricular pressure (e.g., end-systolic pressures (in WT) of 162.0 mmHg and (in KO) of 155.4 mmHg, Table 1), dP/dtmax and -dP/dtmin were significantly reduced, and Tau was increased in KO mice (Table 1). Comparable alterations in these parameters were not observed after chronic AngII administration, indicative of the activation of a distinct intracellular signaling pathway for the distinct hypertrophic stimuli investigated herein.

### 3.10. An Increased Phospho-Phospholamban-to-Phospholamban Ratio in KO Mice

First, we asked if ATIP1 deficiency might be sufficient to change the ratios of phosphorylated signaling molecules or their expressions in the heart. Under basal conditions (untreated mice), no changes in the ratios of phospho-ERK (P-ERK)-to-ERK (example in Figure 7A) and phospho-troponin I (P-Tn I)-to-troponin I (Figure 7B) were observed. In the hearts of the untreated mice, the expressions of the total ERK, total PLB, total CamKII, and total Tn I were found to be unaltered as well (exemplary western blots in Figure 7A,B; in total, n = 12 mice in each group, no statistically significant changes were observed). From the tested markers, only the ratio of the phospho-phospholamban (P-PLB)-to-phospholamban was increased in KO mice as compared to that in wild-type mice (2.01 ± 0.6-fold ratio as compared to WT mice, *p* < 0.05, n = 24 for each genotype, Figure 7B).

### 3.11. Decreased ATIP1 Expression After Chronic AngII Treatment

To determine if the ATIP1 levels change in response to a pathway-specific hypertrophic stimulus, we compared ATIP1 expressions in the hearts of untreated WT mice with ATIP1 expressions in the hearts of WT mice after two weeks of chronic AngII administration via osmotic mini pumps. Chronic AngII treatment resulted in a considerable decrease in ATIP1 expression in WT mice in relation to that of GAPDH (Figure 7C, n = 12 mice in each group, * *p* < 0.05).

### 3.12. Increased ERK Phosphorylation in ATIP1-KO Hearts After Chronic AngII Treatment

Next, we asked if this pathway-specific hypertrophic stimulus might alter the phosphorylation or expression of ERK. The western blots of the ATIP1-KO and WT heart lysates from the mice that had been chronically treated with AngII via osmotic mini pumps for two weeks showed a significant genotype-dependent relative increase in ERK phosphorylation in ATIP1-KO mice (Figure 7D, n = 12 mice in each group, * *p* < 0.05), indicative of a missing anti-hypertrophic inhibition mediated by the new AT2/ATIP-signaling pathway.

### 3.13. Relative Increases in Proteins Involved in Ca^2+^ Cycling and Ca^2+^-Dependent Signaling in ATIP1-KO Hearts After Chronic AngII Treatment

According to the changes in Ca^2+^ cycling and Ca^2+^-dependent signaling, we asked if the AngII-induced reduction in the left-ventricular performance of ATIP-KO hearts might be paralleled by changed expression levels of proteins involved in Ca^2+^ cycling and Ca^2+^-dependent signaling. After chronic AngII administration, the expression levels of SERCA2a and phospholamban were approximately twofold higher compared to those in WT controls, and the expressions of the voltage-gated Ca^2+^ channel Cav_1.2_ and the Ca^2+^/calmodulin-dependent kinase CaMKII increased by about 50% (Figure 7E, n = 12 mice in each group, * *p* < 0.05).

## 4. Discussion

Our aim was to unravel the physiological functions of ATIP1 in a loss-of-function mouse model generated by a gene trap approach. In sum, our results suggest the important roles of ATIP1 in AT2-dependent signaling, in cardiac hypertrophy, and in the regulation of cellular contractility.

Our findings in cardiomyocytes are in line with previous publications demonstrating ATIP1 cellular membrane association but are in sharp contrast to the suggested localization in mitochondria [17,20,21]. According to our data, we conclude that membrane-associated interaction with AT2 is likely. Previous publications [19,21], suggesting the existence of various ATIP1 splice isoforms [19,21], prompted us to identify variants expressed in the heart. Using primers, which cover all the predicted splice variants, we could sufficiently detect the expression of the murine ATIP1 isoform 4 only. However, we cannot completely exclude the existence of other predicted forms or their expressions in other tissues. The functionality and nomenclature of different ATIP1 splice isoforms is not finally set yet. Indeed, different names were used for equivalent variants, e.g., ATIP1, MTUS1, and ATBP50 stand for the shortest variant, originally called Mtus1, variant 4 (NCBI: NM_001005865.2), which we found to be the most abundant in the heart. Additionally, the mouse Mtus1 gene is lacking exons as compared to those in the corresponding human gene, which possibly results in a less complex expression pattern [40]. The genetic organization of the Mtus1 gene and a comprehensive overview of the described splice variants in different species depict the complexity of this topic [40], which needs further clarification in the future.

In our model, the gene trap vector insertion resulted in disruptions not only at the genomic level but also at the RNA and protein levels. There were no signs of leakiness detectable, not even with an antibody generated against the common C-terminus of all the possible variants. Therefore, we excluded the expressions of read-through products or splice variants. According to our RACE data, identifying isoform 4 as the only isoform expressed in the heart, the gene trap vector inserted behind the third used exon, making the expression of a truncated protein unlikely, was also supported by our western-blotting data. A limitation of this model is that the inserted gene trap cassette consists of a large DNA fragment, which might cause unexpected side effects. However, most gene-targeting strategies use gene-targeting vectors with a length of several thousand base pairs, and off-target effects can never be excluded. The use of the β-geo reporter, which was under the control of the endogenous Mtus1 promoter, revealed a surprisingly high intensity compared to those of the other organs, thus reflecting the strong Mtus1 promoter activity in the heart. Although the regulation of ATIP1 expression by PARP-1 had been investigated using PARP-1 knockout mice, it had not been confirmed in vivo up to now [22]. Therefore, our gene trap model, reflecting the endogenous promoter activity, is an ideal tool to investigate ATIP1 expression in vivo.

The first hint suggesting the putative role of ATIP1 in the heart arose from our observation of an increased heart weight/bodyweight ratio in KO mice without any gender relation and, most importantly, was not caused by altered blood pressure. Transgenic mice constitutively overexpressing ATIP1 under the control of the CMV promoter did not have increased blood pressure or any signs of cardiac hypertrophy, which is in contrast to our mouse model of ATIP1 deficiency [24]. The increase in the heart’s weight was consistent with other typical signs of cardiac hypertrophy in our mouse model, such as broadened cardiomyocytes, almost double the amount of cardiac collagen, and reduced chamber volumes, suggesting concentric hypertrophy. To gain more insight into the mechanisms underlying these phenotypic differences and the previously suggested interaction with AT2 [20,21], we investigated AngII-dependent and AT2-mediated inhibitory pathways. Long-term in vivo treatment with AngII via osmotic mini pumps resulted in a significantly reduced maximum volume, an elevated end-systolic pressure, a reduced end-diastolic volume, and a diminished cardiac output in KO mice as compared to those in WT controls. We interpreted these as signs of compensated cardiac hypertrophy. As reflected by the cardiac output, which is elevated after long-term AngII treatment in WT mice but reduced in KO mice, the hypertrophic effects of AngII seem to outbalance the positive inotropic effects in KO hearts. At the molecular level, after long-term AngII treatment, the ratio of phospho-ERK to ERK, which is a known marker activated in cardiac hypertrophy, was highly increased in ATIP1-KO hearts, confirming the lack of the inhibitory effect of ATIP1 on ERK activity in our mice [20]. These data imply that ATIP1 is involved in the anti-hypertrophic signaling of AT2.

In addition to that, our functional data from isolated cardiomyocytes reveal important insights about the AT1 and AT2 pathways in the regulation of cardiomyocyte contractility. First, our data show that ATIP1 is a critical downstream signal in the AT2 pathway in the regulation of cell shortening. Second, our results point to the important, but previously poorly understood, role of AT2 in the regulation of the excitation–contraction coupling. The role of AT1 in the heart has been widely explored in the past and is well established. AT1 mediates the positive inotropic response of AngII in a dose-dependent manner [41], and AT2 has been suggested to antagonize these AT1 effects, although consistent data are missing [42,43]. However, our data clearly support this concept. In our mouse model of ATIP1 deficiency, isolated myocytes reveal, at low frequencies (i.e., 0.5 and 1 Hz), a significantly increased shortening compared to that in WT mice, although the Ca^2+^ transient amplitudes did not show any significant difference. This is consistent with our in vivo data, which show a significantly increased ejection fraction in ATIP1-KO mice. These findings point to the disruption of the AT2 pathway because of ATIP1 deficiency and the loss of the inhibitory effect of AT2 on AT1, which presumably leads to increased contractility in our ATIP1-KO myocytes. For further exploration, we conducted experiments using acute AngII superfusion on isolated cardiomyocytes. Interestingly, although Ang II significantly increased contractility in cells from WT mice, it was reduced in cells from ATIP1-KO mice by Ang II. In cells from WT mice, the increase in contractility, caused by AngII, reflects the well-known positive inotropic effect of AngII. This does not occur in myocytes from ATIP1-KO mice, which might be because of the already maximally activated AT1 pathway in ATIP1-KO myocytes lacking the inhibitory effect of AT2. Nevertheless, this observation needs to be further validated, as we did not address it in our model. To further explore the putatively defective AT2 pathway, we used the AT2 selective agonist CGP42112A in a subset of experiments. As in our experiments with AngII, CGP42112A exerted a positive inotropic response in cardiomyocytes from WT mice, but contractility was reduced in cells from ATIP1-KO mice. This, furthermore, gives evidence for a properly working AT2 signal cascade in WT cells and a disruption of the AT2 pathway in ATIP1-KO cells. Strikingly, throughout our experiments, we did not detect any differences in the Ca^2+^ transient amplitudes between those groups, albeit significant differences in cell shortening occurred. This might, at least, partly be explained by altered myocyte Ca^2+^ sensitivity in our mouse model, as AngII mainly exerts its inotropic effects through altered myocyte Ca^2+^ sensitivity and only in part through altered Ca^2+^ fluxes, which are reflected by the Ca^2+^ transient amplitude [44,45]. Nevertheless, this finding needs further exploration.

Concerning relaxation properties, the cardiomyocytes from the ATIP1-KO mice exhibited a slightly faster relaxation compared to that in cells from the WT mice. Because the function of the Na^+^/Ca^2+^ exchanger (NCX), measured as Ca^2+^ decline following the local application of caffeine, was unchanged, it is suggestive that increased SERCA2a activity is responsible for the faster relaxation in ATIP1-KO cells. This is in line with the observation of the increased SERCA2a expression in ATIP1-KO hearts. Interestingly, AngII accelerated relaxation in cells from WT mice but not in cells from ATIP1-KO mice, which, furthermore, stresses a disturbed AngII pathway in ATIP1-KO mice. Moreover, relaxation was also accelerated by CGP42112A in cells from WT mice but unaffected in cells from ATIP1-KO mice.

Our in vivo data and data from isolated cardiomyocytes clearly support the biochemically demonstrated evidence of the physical and functional interactions of ATIP1 and AT2 [20,21]. In sum, the loss of ATIP1 disrupts an anti-hypertrophic signaling cascade in KO mice, resulting in spontaneous cardiac hypertrophy. Additionally, we provide the first evidence that ATIP1 is a downstream signal in the AT2 pathway, impacting cardiomyocyte contractility presumably through the disinhibition of the AT1 pathway.

## 5. Conclusions

In sum, the loss of ATIP1 disrupts an anti-hypertrophic signaling cascade in KO mice, resulting in spontaneous cardiac hypertrophy. Additionally, we provide evidence that ATIP1 is a downstream signal in the AT2 pathway impacting cardiomyocyte contractility, presumably through the disinhibition of the AT1 pathway.

## Figures and Tables

**Figure 1 cells-14-00645-f001:**
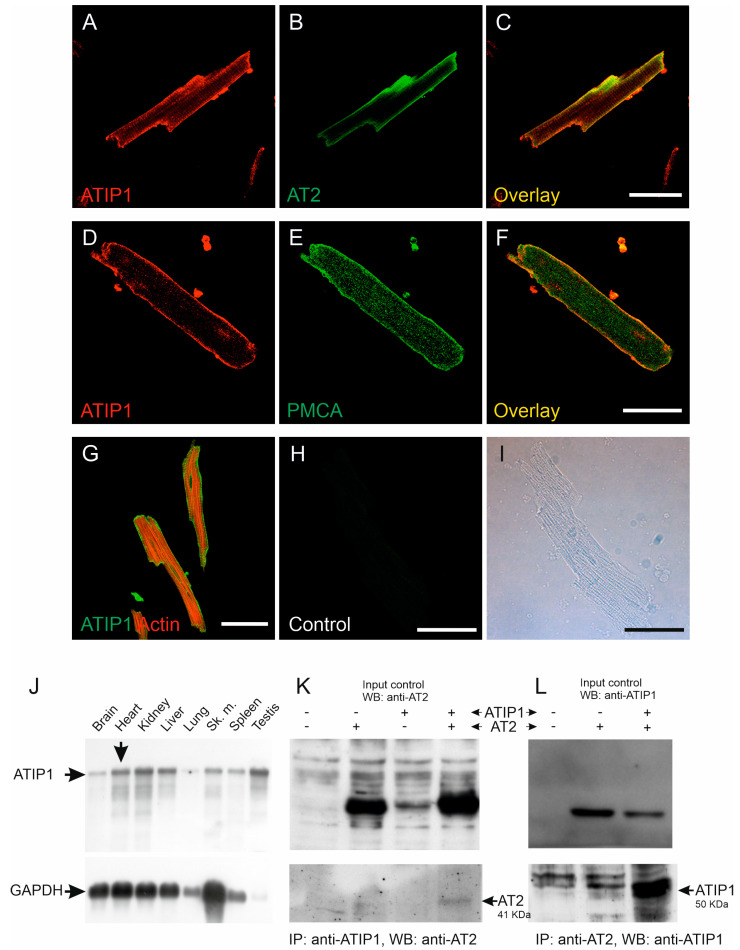
Expression and localization of ATIP1 in the heart. Localization of ATIP1 at the plasma membrane of isolated cardiomyocytes (**A**–**F**), supported by co-localization with AT2 (**B**,**C**) and the plasma membrane’s Ca^2+^ ATPase, PMCA (**E**,**F**) and not with actin (**G**). (**H**,**I**) Antibody controls; bars: 50 μm. (**J**) Expression of ATIP1 in different tissues, detected using northern blotting (BALB/c male mouse, mRNA). In addition to the observed expression in the heart, ATIP1 transcripts were also detected in other tissues, e.g., the kidneys, liver, and testes. (**K**,**L**) Lysates of HEK293 cells, which were single- and double-transfected with ATIP1 and AT2 expression constructs, were immunoprecipitated (IP) with antibodies against ATIP1 (**K**) or AT2 (**L**), and the precipitated material was analyzed using western blotting (WB) with antibodies against AT2 (**K**) and ATIP1 (**L**).

**Figure 2 cells-14-00645-f002:**
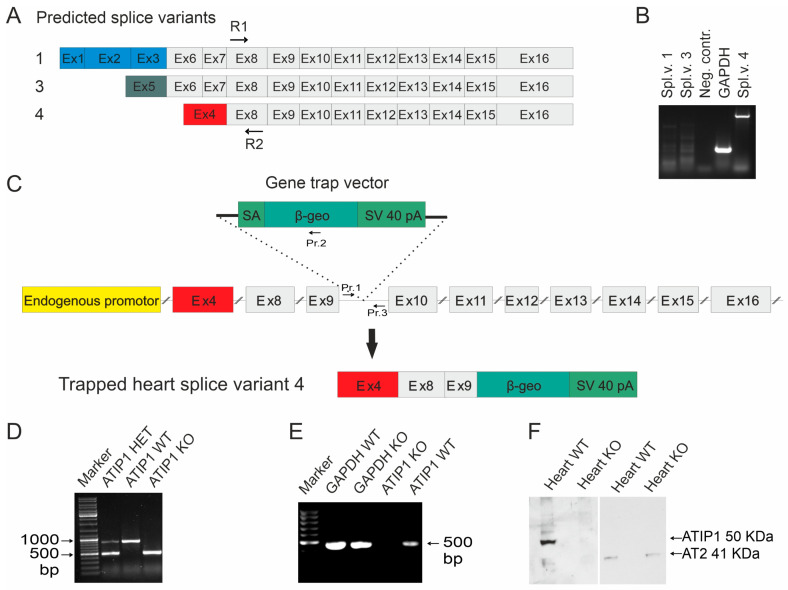
Gene disruption. (**A**) Predicted ATIP1 splice variants and the cloning of heart-specific murine splice variant 4 by RACE-PCR with gene-specific primers R1 and R2 located within common exon 8. Additionally, RT-PCRs for full-length splice variants detected only splice variant 4 but neither variant 1 nor variant 3 in the heart (**B**). (**C**) Insertion of the gene trap vector into the Atip1 gene. The gene trap vector was integrated after the third exon of heart-specific splice variant 4. It contains a splice acceptor (SA), a reporter gene (β-geo), and an SV40 polyadenylation signal. The insertion of the gene trap vector results in artificially spliced fusion transcripts of upstream exons and the β-geo reporter gene (C). The primers (Pr. 1, Pr. 2, and Pr. 3) used for genotyping are shown in (**C**). Genotyping by PCR using primers 1 and 2 for the KO PCR and primers 1 and 3 for the WT PCR. PCR products are 500 bp for KO and 900 bp for WT alleles, respectively (**D**). (**E**) RT-PCR with heart RNA of KO and WT mice verified the knockout at the RNA level. (**F**) Western blots revealed a complete loss of ATIP1 in the hearts of the ATIP1-KO animals, whereas the AT2 expression remained unchanged.

**Figure 3 cells-14-00645-f003:**
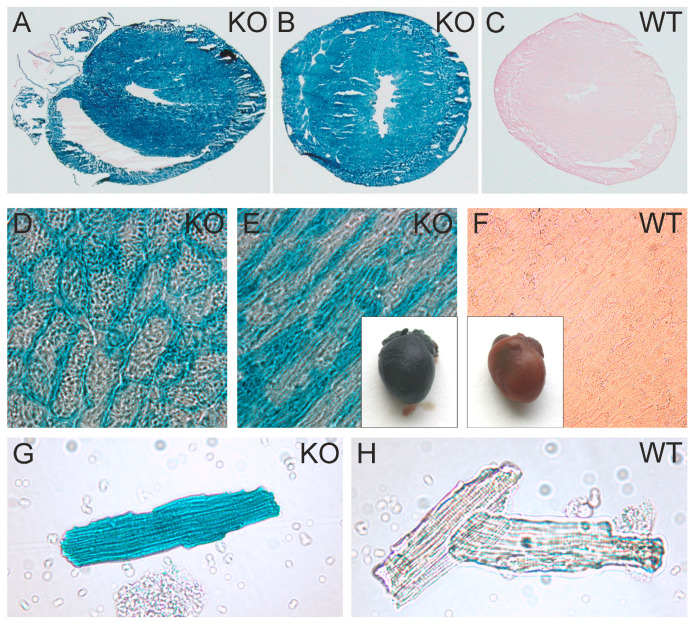
ATIP1 gene promoter activity in the heart and in isolated cardiomyocytes. Intensive X-Gal staining in ATIP1-KO hearts (**E**), cross (**B**,**D**) and longitudinal (**A**,**E**) cryosections, and in isolated cardiomyocytes (**G**) compared to those in WT (**C**,**F**,**H**), which were indicative of strong ATIP1 gene promoter activity in cardiomyocytes.

**Figure 4 cells-14-00645-f004:**
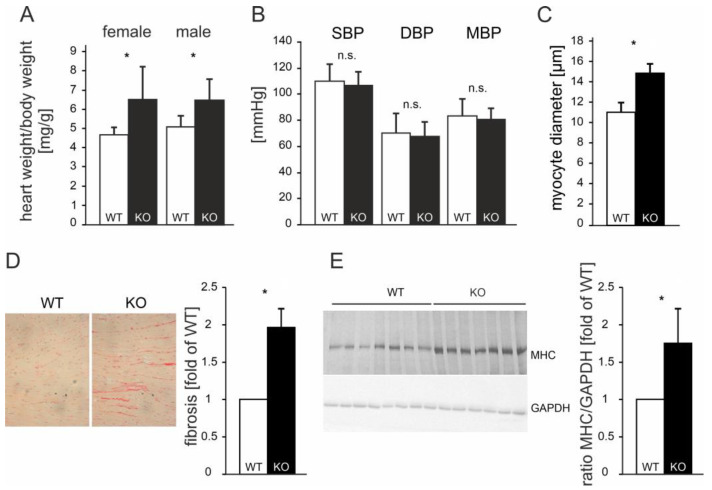
Cardiac hypertrophy in ATIP1-KO mice was not caused by hypertension. The ATIP1-knockout mice spontaneously developed cardiac hypertrophy, as reflected by a significantly increased heart-to-bodyweight ratio from 4.7 mg/g in WT mice to 6.4 mg/g in KO animals (n = 20, * *p* < 0.005, (**A**)) in the presence of an unchanged systemic blood pressure (n = 25, n.s., (**B**)). The hypertrophy was characterized by enlarged diameters of cardiac myocytes, as estimated using quantitative morphometry of PAS-stained left ventricular myocardial sections (n = 100 cells in each group, in sections of 12 mice in each group, * *p* = 2.6 × 10^−5^, (**C**)). (**D**) Relative myocardial fibrosis assessed using picrosirius red staining for collagen was also significantly increased in KO animals, as compared to WT controls (n = 12 mice in each group, * *p* = 0.028). (**E**) Exemplary myosin-heavy-chain (MHC) expression in KO mice as compared to WT controls (left panel) and the quantification of the MHC-to-GAPDH ratio (right panel, n = 12 mice of each genotype, * *p* < 0.05); n.s., not significant.

**Figure 5 cells-14-00645-f005:**
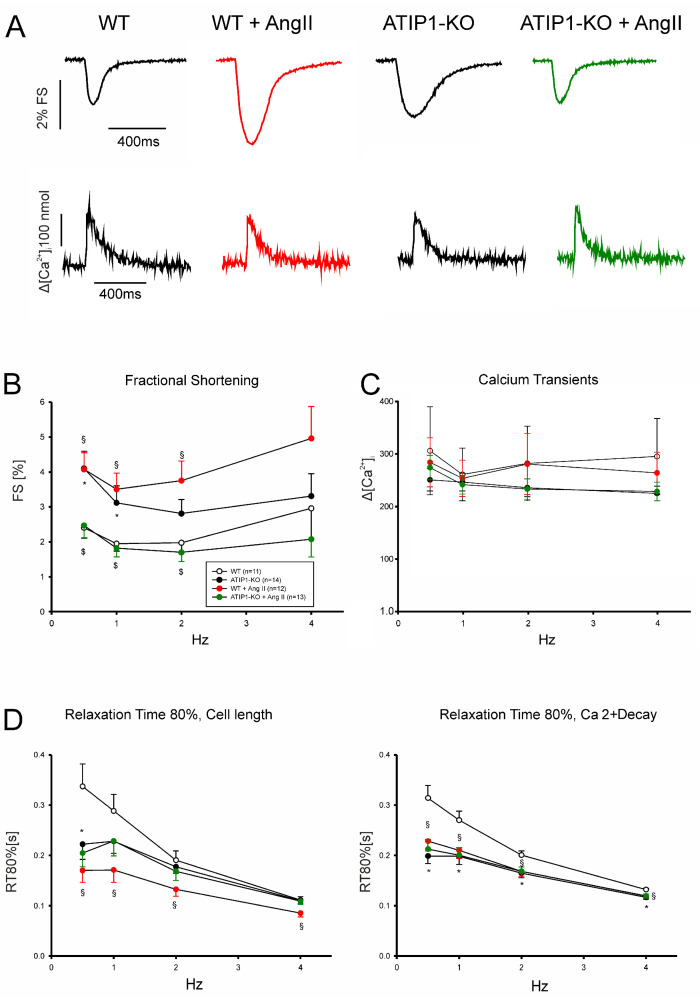
Functional measurements of isolated ventricular cardiomyocytes. (**A**) Exemplary original recordings of fractional cell shortening (upper panels) and Ca^2+^ transient (lower panels) measurements under basal conditions (at 1 Hz) and after treatment with AngII. (**B**) The force–frequency relationship of fractional shortening. Cells from ATIP1-KO (n = 14), WT (n = 11), and ATIP1-KO treated with AngII (n = 13) and WT plus AngII (n = 12) mice; §: *p* < 0.05 for WT vs. WT + AngII; $: *p* < 0.05 for ATIP1-KO vs. ATIP1-KO + AngII; and (**C**) Ca^2+^ transient amplitude. (**D**) Relaxation is measured as the cell length relaxation to 80% of the resting cell length and Ca^2+^ decline; §: *p* < 0.05 for WT vs. WT + AngII; *: *p* < 0.05 WT + AngII vs. KO + AngII.

**Figure 6 cells-14-00645-f006:**
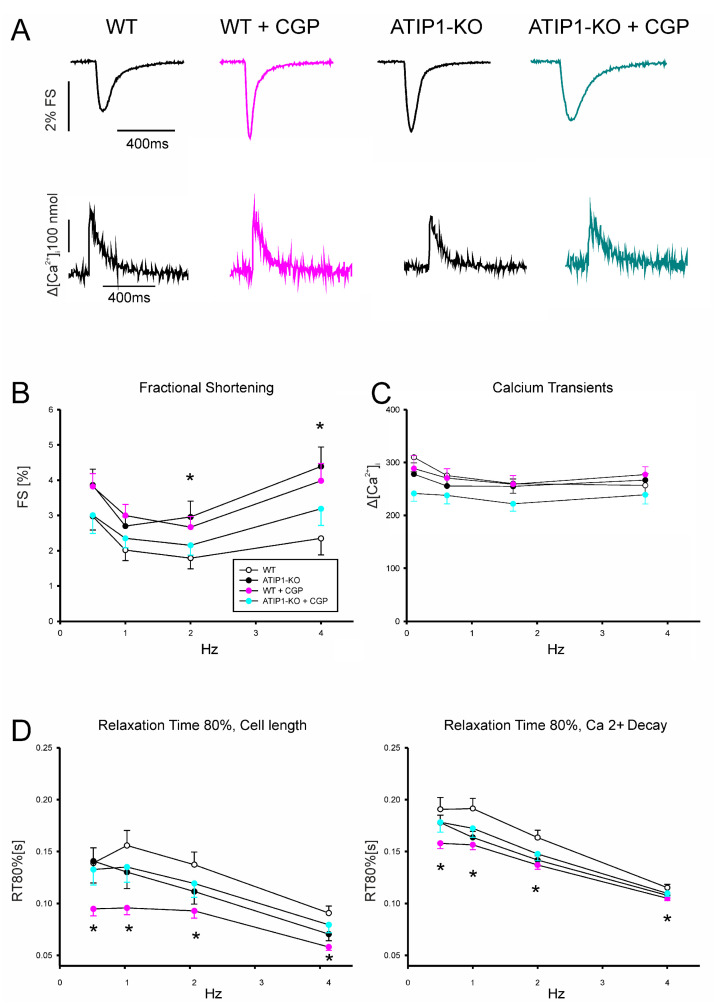
(**A**) Original recordings of fractional cell shortening (upper panels) and Ca^2+^ transient (lower panels) measurements under basal conditions and after treatment with AT2 agonist CGP. (**B**) The force–frequency relationship for fractional shortening. Cells from ATIP1-KO (n = 9), WT (n = 7), ATIP1-KO treated with CGP (n = 8), and WT plus CGP (n = 7) mice and (**C**) Ca^2+^ transient amplitude. (**D**) Relaxation is measured as the cell length relaxation to 80% of the resting cell length and Ca^2+^ decline; * *p* < 0.05 for WT vs. WT + CGP.

**Figure 7 cells-14-00645-f007:**
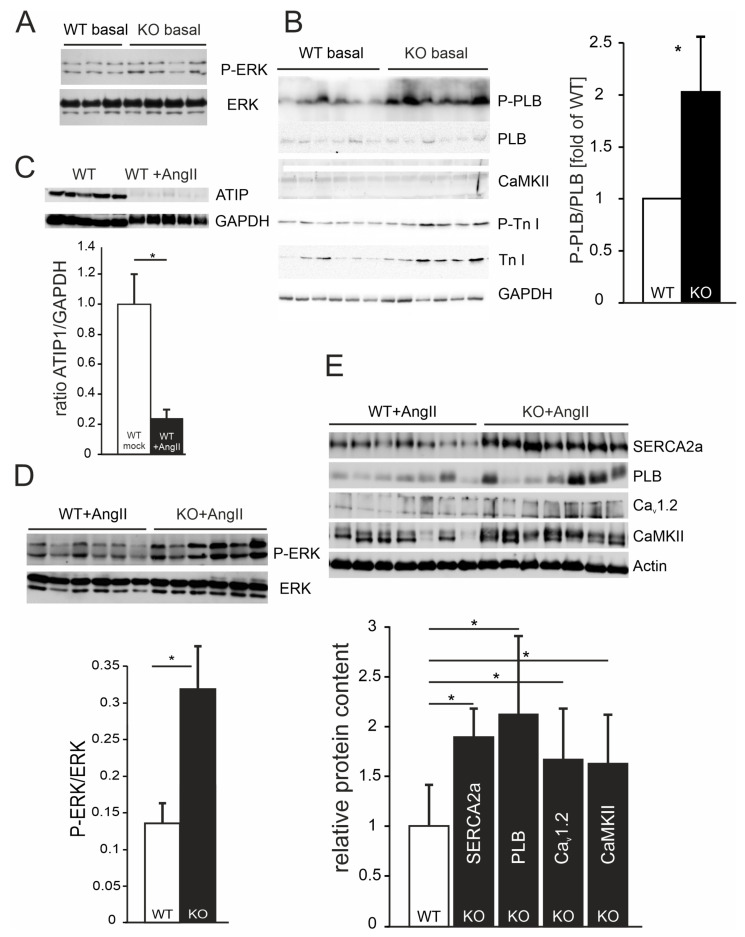
(**A**) Exemplary western blots showing no differences in phospho-ERK or ERK expressions in the hearts of untreated (basal) KO mice as compared to those in the hearts of untreated WT mice. (**B**) Western blots using heart lysates of untreated WT and KO mice. P-PLB, anti-phopho-phospholamban; PLB, phospholamban; CaMKII, Ca^2+^/calmodulin-dependent protein kinase II; P-Tn I, phospho-troponin I; Tn I, troponin I. Statistically significant differences were detected for the ratio of P-PLB/PLB (* *p* < 0.05; n = 24 mice in each group), all the other ratios were not changed between WT and KO basal mice as compared to GAPDH (n = 12 mice in each group). (**C**) Exemplary western blots showing ATIP1 expressions in WT heart lysates, either untreated or after chronic AngII administration in relation to GAPDH. The quantification of further blots revealed a decrease in ATIP1 expression to about 20% of the levels expressed in untreated WT mice (n = 12 mice in each group, * *p* < 0.05). (**D**) Western blot analysis revealed a significant increase in ERK phosphorylation in the hearts of ATIP1-KO animals after two weeks of chronic AngII infusion (n = 12 mice in each group, * *p* < 0.05). (**E**) Relative protein expressions of SERCA2a, PLB, voltage-gated Ca^2+^ channel (Ca_v_1.2), and CaMKII. All the signals were normalized to cardiac alpha-actin, WT expression levels were set at a value of “1”, and KO signals are shown in relation to the WT value (n = 12 mice in each group, * *p* < 0.05).

**Table 1 cells-14-00645-t001:** Hemodynamic parameters before and after chronic AngII administration or transversal aortic constriction (TAC), n = 12 in each group. ^A^: *p* ≤ 0.05 for KO without Ang II vs. WT without Ang II; ^B^: *p* ≤ 0.05 for WT after Ang II vs. WT without Ang II; ^C^: *p* ≤ 0.05 for KO after Ang II vs. KO without Ang II; ^D^: *p* ≤ 0.05 for KO after Ang II vs. WT after AngII; ^E^: *p* ≤ 0.05 for KO TAC vs. WT TAC; NIBP, non-invasive blood pressure; HW/BW, heart weight/bodyweight ratio; CO, cardiac output; dP/dtmax, the peak rate of the pressure rise; dP/dtmin, the peak rate of the pressure decline; Ea, arterial elastance (a measure of the ventricular afterload); EDV, end-diastolic volume; EF, ejection fraction; ESP, end-systolic pressure; ESV, end-systolic volume; HR, heart rate; Tau (W), relaxation time constant calculated using the Weiss method (the regression of log (pressure)); V@dP/dt max, the volume at dP/dt max; V@dP/dt min, the volume at dP/dt min; P@dP/dt max, the pressure at dP/dt max; SV, stroke volume.

**Hemodynamics/NIBP**	**Untreated**		**After Ang II**		**TAC**	
	**WT**	**KO**	**WT**	**KO**		
**NIBP**						
SBP (mmHg)	112	106	129 ^B^	130 ^C^		
DBP (mmHg)	75	71	90	86		
MBP (mmHg)	87	84	102	102		
HR (min^−1^)	678	620	617	592		
**Hemodynamic parameter** **(isoflurane)**					**WT**	**KO**
HW/BW (mg/g)	4.7	6.4 ^A^	5.0	6.6 ^D^	6.9	7.6 ^E^
HR (min^−1^)	478.00	508.36	489.56	446.89 ^C^	505.1	493.90
Maximum Volume (µL)	23.08	17.32 ^A^	14.49 ^B^	11.63 ^C,D^	21.2	20.6
Minimum Volume (µL)	15.06	9.65 ^A^	5.33 ^B^	3.67 ^C,D^	10.5	9.5
Maximum Pressure (mmHg)	100.36	96.92	105.51	108.15 ^C^	178.9	165.1
SV (µL)	8.02	7.68	5.32	7.45	10.6	8.2 ^E^
ESP (mmHg)	99.03	93.27	100.49	103.01 ^C^	162.0	155.4
ESV (µL)	13.58	10.44 ^A^	5.55 ^B^	4.23 ^C^	12.2	14.0
EDV (µL)	22.34	16.45 ^A^	13.72 ^B^	10.42 ^C,D^	20.8	20.0
CO (µL min^−1^)	3844.92	3941.22	4445.77	3519.55 ^D^	5345.4	3829.0 ^E^
V@dP/dt max (µL)	22.60	16.88 ^A^	13.65 ^B^	11.12 ^C,D^	20.6	20.1
V@dP/dt min (µL)	15.35	10.25 ^A^	5.61 ^B^	4.20 ^C^	11.1	12.9
P@dP/dt max (mmHg)	53.48	53.87	61.09	64.85 ^C,D^	63.9	57.9
Ea (mmHg µL^−1^)	13.32	12.72	11.54	13.65 ^C,D^	15.6	20.4 ^E^
**Systolic indices**						
EF (%)	36.48	45.85 ^A^	63.37 ^B^	69.81 ^C^	53.8	47.0
dP/dt_max_ (mmHg s^−1^)	6966.56	8217.14 ^A^	8823.00	9448.33 ^C^	10,466.2	8464.4 ^E^
dV/dt max (µL s^−1^)	37,256	32,986	36,525	28,625 ^D^	725.2	627.1
**Diastolic indices**						
−dP/dt_min_ (mmHg s^−1^)	6806.33	8364.59 ^A^	9060.78 ^B^	8855.67	10,624.1	8383.6 ^E^
Tau (W) (ms)	7.66	5.92 ^A^	6.48	6.55	6.6	8.3 ^E^

## Data Availability

The original contributions presented in this study are included in the article. Further inquiries can be directed to the corresponding author(s).

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
