# Peer review of "ATIP1 Is a Suppressor of Cardiac Hypertrophy and Modulates AT2-Dependent Signaling in Cardiac Myocytes"

_cells, 2025, doi:10.3390/cells14090645_

Round 1

Reviewer 1 Report

Comments and Suggestions for Authors

This study presents novel insights into the role of ATIP1 in cardiac hypertrophy and AT2 receptor signaling, providing strong evidence that ATIP1 deficiency leads to spontaneous cardiac hypertrophy and altered cardiomyocyte contractility. The study establishes a unique connection between ATIP1 and anti-hypertrophic signaling via the AT2 receptor, contributing to a deeper understanding of AngII-driven cardiac remodeling. Overall, this study makes a significant contribution to understanding ATIP1’s role in cardiac physiology. The novelty lies in demonstrating that ATIP1 deficiency disrupts AT2-dependent signaling, leading to hypertrophy and altered contractility. However, addressing several concerns regarding data presentation, clarity, and logical consistency would significantly enhance the manuscript’s quality.

Major Concerns & Suggested Revisions

  1. Strengthening the Introduction with References and Clinical Relevance

The introduction should include references that contextualize how this study extends previous knowledge of ATIP1’s role in cardiovascular physiology. Additionally, the authors should discuss the potential clinical relevance of ATIP1—for example, how its role in hypertrophic cardiomyopathy, heart failure, or therapeutic targeting could influence cardiovascular treatments. This would improve the translational significance of the study.

  1. The terms must be consistent throughout the manuscript.

In line 297, "Atip1" should be corrected to "ATIP1" to maintain uniformity.

The authors should clarify whether "ATIP1" refers to the protein and "Atip1" to the gene. If so, this distinction should be clearly stated to avoid confusion.

  1. Western Blot Data for ATIP1 Expression in the Kidney (Figure 2F)

In Figure 2F, the authors do not include Western blot data for ATIP1 expression in the kidney, even though Figure 7B examines renin activity, which is kidney-related. Since the kidney plays a critical role in ATIP1 function, including Western blot data for kidney expression would strengthen the link between Figures 2F and 7B. If these data are unavailable, the authors should explicitly justify their omission and discuss its implications.

  1. Reorganization and Clarification of Figure 7

Figure 7 is a key result but lacks a clear logical structure, making it difficult to interpret. Figure 7A presents WT, WT+AngII, and WT+TAC groups for ATIP1 expression analysis. Figure 7B presents WT and KO groups with or without TAC to analyze renin activity. Figures 7C and 7D compare WT+AngII and KO+AngII but ignore the WT control for key markers like p-ERK and SERCA2a/PLB, making interpretation incomplete. Therefor suggest to focus on Figures 7C and 7D, which provide the most relevant Western blot data. Remove Figures 7A and 7B, as they introduce additional variables (TAC, renin activity) that do not directly support the main mechanistic conclusion. Ensure that Figures 7C and 7D include WT controls for p-ERK and SERCA2a/PLB to enhance clarity.

Terminology Consistency in Figure 7B ("Aortic Constriction" vs. "TAC"): The text in Figure 7B refers to "aortic constriction", while the figure label states "TAC". The authors should clarify whether these terms refer to the same experimental procedure to avoid confusion.

Final Recommendation: This study presents important insights into ATIP1’s role in cardiac hypertrophy and AT2 signaling, but several key revisions are necessary to improve clarity, consistency, and logical presentation. Addressing these concerns will significantly enhance the manuscript’s scientific rigor and impact. If these revisions are incorporated, the study will be suitable for publication and provide meaningful contributions to the field of cardiovascular research.

Comments on the Quality of English Language

as the Comments and Suggestions for Authors

Author Response

Point-by-Point Reply

 Reviewer 1:

Comments and Suggestions for Authors

This study presents novel insights into the role of ATIP1 in cardiac hypertrophy and AT2 receptor signaling, providing strong evidence that ATIP1 deficiency leads to spontaneous cardiac hypertrophy and altered cardiomyocyte contractility. The study establishes a unique connection between ATIP1 and anti-hypertrophic signaling via the AT2 receptor, contributing to a deeper understanding of AngII-driven cardiac remodeling. Overall, this study makes a significant contribution to understanding ATIP1’s role in cardiac physiology. The novelty lies in demonstrating that ATIP1 deficiency disrupts AT2-dependent signaling, leading to hypertrophy and altered contractility. However, addressing several concerns regarding data presentation, clarity, and logical consistency would significantly enhance the manuscript’s quality.

Major Concerns & Suggested Revisions

  1. Strengthening the Introduction with References and Clinical Relevance

The introduction should include references that contextualize how this study extends previous knowledge of ATIP1’s role in cardiovascular physiology. Additionally, the authors should discuss the potential clinical relevance of ATIP1—for example, how its role in hypertrophic cardiomyopathy, heart failure, or therapeutic targeting could influence cardiovascular treatments. This would improve the translational significance of the study.

Reply: We fully agree with this point and added a paragraph (lines 76 – 83) in the introduction and three additional references [# 25,26,27] describing either the role of ATIP1 hypertrophic cardiomyopathy in different mouse models and the role of ATIP1 in non-compaction of ventricular mayocardium in humans

  1. The terms must be consistent throughout the manuscript.

In line 297, "Atip1" should be corrected to "ATIP1" to maintain uniformity.

The authors should clarify whether "ATIP1" refers to the protein and "Atip1" to the gene. If so, this distinction should be clearly stated to avoid confusion.

Reply: To maintain uniformity we have changed the format to ATIP1 throughout the manuscript and in case it refers to the gene, we have added the word “gene”.

  1. Western Blot Data for ATIP1 Expression in the Kidney (Figure 2F) and
  2. Reorganization and Clarification of Figure 7

In Figure 2F, the authors do not include Western blot data for ATIP1 expression in the kidney, even though Figure 7B examines renin activity, which is kidney-related. Since the kidney plays a critical role in ATIP1 function, including Western blot data for kidney expression would strengthen the link between Figures 2F and 7B. If these data are unavailable, the authors should explicitly justify their omission and discuss its implications.

Figure 7 is a key result but lacks a clear logical structure, making it difficult to interpret. Figure 7A presents WT, WT+AngII, and WT+TAC groups for ATIP1 expression analysis. Figure 7B presents WT and KO groups with or without TAC to analyze renin activity. Figures 7C and 7D compare WT+AngII and KO+AngII but ignore the WT control for key markers like p-ERK and SERCA2a/PLB, making interpretation incomplete. Therefor suggest to focus on Figures 7C and 7D, which provide the most relevant Western blot data. Remove Figures 7A and 7B, as they introduce additional variables (TAC, renin activity) that do not directly support the main mechanistic conclusion. Ensure that Figures 7C and 7D include WT controls for p-ERK and SERCA2a/PLB to enhance clarity.

Reply: We apologize for this confusion we might have generated with these two Figures. As suggested, and to focus on the main mechanistic conclusion, we have restructured the Figures 2 and 7 and put the focus on the potential mechanisms in the heart. Although ATIP1 might play a role in kidney function, we have not added a Western demonstrating the loss of ATIP1 in the kidney, since Reviewer 1 also suggested to delete the original Figure 7B displaying Renin activity. Instead, we have included Western blots showing loss of ATIP1 only in the heart of KO mice despite normal AT2 expression (Figure 2F) and reorganized Figure7 accordingly.

In the new Figure 7, we have included data of untreated WT and KO control mice, and we have tried to put our focus on p-ERK and Ca2+ handling proteins. In (A), exemplary p-ERK/ERK Westerns of WT and KO hearts of untreated mice were included. In (B), exemplary Western blots for PLB, CaMKII, and Troponin were included, however, none of these blots revealed significant differences, which brought us to the conclusion that these protein levels remain more or less unchanged in untreated mice. In the new Figure 7C (originally 7B), we have put our focus on the AngII-induced data (and removed TAC-related data), because it guides to Figures 7D and E, demonstrating significant changes in displayed protein levels. Unfortunately, it was not possible to include WT basal data in the Figures 7D and E (as suggested by the Reviewer), because this would require a new series of experiments with or without implantation of osmotic AngII mini pumps, which is impossible in the time frame of this paper revision. Hence, we decided to include the basal data in Figure 7 A and B.

Terminology Consistency in Figure 7B ("Aortic Constriction" vs. "TAC"): The text in Figure 7B refers to "aortic constriction", while the figure label states "TAC". The authors should clarify whether these terms refer to the same experimental procedure to avoid confusion.

Reply: The original Figure 7B was removed. Therefore, TAC and aortic constriction were deleted, as well.

Final Recommendation: This study presents important insights into ATIP1’s role in cardiac hypertrophy and AT2 signaling, but several key revisions are necessary to improve clarity, consistency, and logical presentation. Addressing these concerns will significantly enhance the manuscript’s scientific rigor and impact. If these revisions are incorporated, the study will be suitable for publication and provide meaningful contributions to the field of cardiovascular research.

Reply: Thank You!

Reviewer 2 Report

Comments and Suggestions for Authors

In this manuscript, the authors tried to evaluate the physiological function of ATIP1 in the heart using an ATIP1 knockout mouse model and isolated primary cardiomyocytes. Here are several points that could help to improve the quality of the manuscript.

-The problem of the mouse model: the authors used gene-trapping techniques to establish the ATIP1 knockout mouse model which inserts a large fragment in the mouse genome and could potentially cause some unexpected effects. It’s almost impossible to use an alternative knockout strategy in the revision period, so at least this point should be discussed as a limitation of the study.

-The authors stated this mouse model as the first description of ATIP1-deficiency on cardiac function, however, a group has reported the cardiac hypertrophic phenotype in MTUS1 (ATIP1) knockout mice using a similar strategy in 2012 (PMID: 22200760). The authors should carefully read through this article and discuss the conclusions of the reference.

-For all the phenotype evaluations, the age of the mice is missing. Did the authors evaluate the development of cardiac hypertrophy as the mice age? As the authors observed the increased cardiomyocyte size and enhanced fibrosis, will the mice eventually develop heart failure?

-It’s confusing to observe almost no response to AngII in ATIP1-KO cardiomyocytes in Figure 5. In the molecular mechanism part, the authors showed that ATIP1-KO leads to enhanced downstream signaling upon AngII infusion, supporting ATIP1-AT2 as a suppressor of AngII signaling, and ATIP1-KO should derepress this effect and show a stronger response upon AngII treatment. Did the authors have any explanation on this point and this should also be discussed.

-Some technique issues: 1) For IP assays, the blot for bait protein and input should be provided as the control and MW or the size of the marker is missing in all western blot, IP-WB, and northern blot assays. 2) For the X-Gal assay, the figure is labeled as KO and the main text indicated heterozygous, so it’s better to label it clearly in the figure. Moreover, did the author determine the LacZ expression in other organs besides the heart? 3) For Figure 4C, the authors use myocyte diameter to indicate the hypertrophy. Why didn’t the authors use the cardiomyocyte area to reveal the size of the cardiomyocytes? How was the myocyte diameter measured? 4) For Figure 4, did the authors measure the marker gene expression of cardiac hypertrophy and fibrosis? 5) The N number should be labeled in each group. 6) For the downstream signaling, did the authors observe the change of p-ERK at baseline as the hypertrophy happens without any stimulations?

Author Response

Reviewer 2

Comments and Suggestions for Authors

In this manuscript, the authors tried to evaluate the physiological function of ATIP1 in the heart using an ATIP1 knockout mouse model and isolated primary cardiomyocytes. Here are several points that could help to improve the quality of the manuscript.

-The problem of the mouse model: the authors used gene-trapping techniques to establish the ATIP1 knockout mouse model which inserts a large fragment in the mouse genome and could potentially cause some unexpected effects. It’s almost impossible to use an alternative knockout strategy in the revision period, so at least this point should be discussed as a limitation of the study.

Reply: We agree with Reviewer 2 that the gene trap cassette comprises of a larger DNA fragment but this is the case for most gene targeting strategies. In most strategies, a neomycin resistance gene (for stem cell selection) and often a reporter (for expression studies) is inserted into the genome, thereby disrupting the targeted genetic structure. In most cases, genetic side effects cannot be excluded and it has to be tested if the gene targeting results in a complete loss-of-function of the targeted protein. We could show this on a genomic level (Figure 2D), RNA level (Figure 2E), and protein level (Figure 2F). Furthermore, beta-Gal expression points towards a correct splicing of the gene trap, because beta-Gal is only expressed if the splice acceptor (SA) of the gene trap vector is used instead of the splice acceptor of the following endogenous exon (Figure 2C). We inserted a sentence on this in the discussion section (lines 550 – 553).

-The authors stated this mouse model as the first description of ATIP1-deficiency on cardiac function, however, a group has reported the cardiac hypertrophic phenotype in MTUS1 (ATIP1) knockout mice using a similar strategy in 2012 (PMID: 22200760). The authors should carefully read through this article and discuss the conclusions of the reference.

Reply: Reviewer 2 is correct, in this paper a similar model, originating most likely from the same stem cell line was used, but no functional or mechanistic data was provided and no possible link between the cardiac hypertrophy and the observed lymphoproliferative disease was investigated. In our model, we did not observe a lymphoproliferative disease, therefore, we focused on the cardiac function of ATIP1. We included the reference [25) in the introduction (lines 76 – 82).

-For all the phenotype evaluations, the age of the mice is missing. Did the authors evaluate the development of cardiac hypertrophy as the mice age? As the authors observed the increased cardiomyocyte size and enhanced fibrosis, will the mice eventually develop heart failure?

Replay: Throughout this study, adult ATIP1-KO mice (6 – 10 months of age) with a biallelic knockout of ATIP1 and wild-type littermate control mice were used. To avoid inbreed effects resulting from the genetic background, we used mice with a mixed 129/Ola×C57Bl/6 genetic background (included in lines 98 – 101). We did not see signs of cardiac failure, therefore we interpreted this as a compensated cardiac hypertrophy (line 378).

-It’s confusing to observe almost no response to AngII in ATIP1-KO cardiomyocytes in Figure 5. In the molecular mechanism part, the authors showed that ATIP1-KO leads to enhanced downstream signaling upon AngII infusion, supporting ATIP1-AT2 as a suppressor of AngII signaling, and ATIP1-KO should derepress this effect and show a stronger response upon AngII treatment. Did the authors have any explanation on this point and this should also be discussed.

Reply: Unfortunately, we do not have a stringent explanation for this. As shown in Figure 5, some investigated parameters were already changed without AngII administration, maybe there were already compensatory mechanisms activated in the KO myocytes without AngII treatment. We have included a small paragraph in the discussion section, lines 600 – 604.

-Some technique issues: 1) For IP assays, the blot for bait protein and input should be provided as the control and MW or the size of the marker is missing in all western blot, IP-WB, and northern blot assays.

The input/transfection controls are included in Figure 1.

To give a better overview, the blots were reduced to minimal size but the original blots are now uploaded as well, if possible marker sizes are included in these uploaded files.

2) For the X-Gal assay, the figure is labeled as KO and the main text indicated heterozygous, so it’s better to label it clearly in the figure. Moreover, did the author determine the LacZ expression in other organs besides the heart?

Reply: We are very sorry for this mistake! Of course, as indicated above, only homozygous KO mice were used in this study.

Of course, we did. But as suggested by reviewer 1, we focused on the heart, therefore we did not include further X-Gal stainings.

3) For Figure 4C, the authors use myocyte diameter to indicate the hypertrophy. Why didn’t the authors use the cardiomyocyte area to reveal the size of the cardiomyocytes? How was the myocyte diameter measured?

Obviously, our methodological description was a little bit short. We have extended this part (lines 209 – 213) and we hope that is more concise now.

4) For Figure 4, did the authors measure the marker gene expression of cardiac hypertrophy and fibrosis?

Reply: For histological analysis, hearts were fixed in 4% formaldehyde and embedded in paraffin. 5 μm sections were stained with periodic acid Schiff or 0.1% picrosirius red. Photomicrographs of myocardial sections were photographed with a CCD-FV2T camera (Olympus) and analyzed with the CellD image analysis system (Olympus), with the investigator blinded to the genotypes (lines 205 – 209).

5) The N number should be labeled in each group.

Reply: Now included in Figure legends.

6) For the downstream signaling, did the authors observe the change of p-ERK at baseline as the hypertrophy happens without any stimulations?

Reply: Yes, now included in Figure 7 A,B and in lines 414 - 422.

Round 2

Reviewer 1 Report

Comments and Suggestions for Authors

The authors have addressed all my questions, and I have no further inquiries.

Author Response

Thank you!

Reviewer 2 Report

Comments and Suggestions for Authors

The authors answered most of my questions and two are remaining here.

-For the IP assays in Fig 1K-L, it’s required to show the WB for bait protein and target protein for all the samples as input control, and the WB of bait protein is also required for IP samples to validate the effectiveness of the IP assays.

- For Figure 4, the authors want to conclude the change of hypertrophy and fibrosis. Only the results from staining are not sufficient to prove the conclusions. At least, the authors need to measure the marker gene expression of cardiac hypertrophy and fibrosis.

Author Response

The authors answered most of my questions and two are remaining here.

-For the IP assays in Fig 1K-L, it’s required to show the WB for bait protein and target protein for all the samples as input control, and the WB of bait protein is also required for IP samples to validate the effectiveness of the IP assays.

Reply: Of course, we checked the functionality of our expression constructs in input lysates prior to the immunoprecipitation assays. We have added new Westerns of input controls in the same order as in the IP Westerns. However, I do not agree with the Reviewer that such a comparison of signal strength allows a validation of effectiveness of the IP assays. In our case, four different antibodies (IPs, Western blots, first and second Abs) were used, all of them with different affinities for their specific antigen. Therefore, it cannot be concluded, if a weak signal in the Western of the IP results from the precipitating or the detecting antibody or even the second antibody or – in the worst case – from a weak detection reagent.

- For Figure 4, the authors want to conclude the change of hypertrophy and fibrosis. Only the results from staining are not sufficient to prove the conclusions. At least, the authors need to measure the marker gene expression of cardiac hypertrophy and fibrosis.

Reply: We agree with the Reviewer that, in addition to the quantification of fibrosis, some marker gene/protein levels of hypertrophy-inducing signal pathways might be changed. To address this point, we retested some of the markers already displayed in Figure 7 with a larger number of WT and KO mice (n=24 each group) and found in this larger cohort the ratio of P-PLB/PLB significantly increased in KO mice. Furthermore, we quantified Myosin heavy chain levels, which were also increased in KO mice, data now included in Figure 4. We did not include further “neutral” data of tested but unchanged markers, e.g. Caspase-1, Atg5, P-p38/p38.